# Error Analysis of a Spherical Capacitive Sensor for the Micro-Clearance Detection in Spherical Joints

**DOI:** 10.3390/mi11090837

**Published:** 2020-09-03

**Authors:** Wen Wang, Wenjun Qiu, He Yang, Keqing Lu, Zhanfeng Chen, Bingfeng Ju

**Affiliations:** 1School of Mechanical Engineering, Hangzhou Dianzi University, Hangzhou 310018, China; wangwn@hdu.edu.cn (W.W.); qwjhdu@163.com (W.Q.); lkq@hdu.edu.cn (K.L.); czf@hdu.edu.cn (Z.C.); 2School of Mechanical Engineering, Zhejiang University, Hangzhou 310027, China; mbfju@zju.edu.cn; 3State Key Lab of Fluid Power & Mechatronic Systems, Zhejiang University, Hangzhou 310027, China

**Keywords:** spherical joint, clearance measurement, capacitive sensor, error analysis

## Abstract

Spherical joints have attracted increasing interest in the engineering applications of machine tools, industrial robots, medical equipment, and so on. As one of the promising methods of detecting the micro-clearance in spherical joints, the measurement accuracy of a spherical capacitive sensor could be affected by imperfectness during the manufacturing and installation of the sensor. This work presents error analysis of a spherical capacitive sensor with a differential structure and explores the dependence of the differential capacitance on manufacturing and the installation imperfectness. Five error sources are examined: the shape of the ball and the capacitive plate, the axial and radial offset of the plate, and the inclined installation of the plate. The mathematical models for calculating the capacitance errors of the spherical capacitive sensor are deduced and validated through a simulation using Ansoft Maxwell. The results show that the measurement accuracy of the spherical capacitive sensor is significantly affected by the shape of plates and ball, the axial offset, and the inclined angle of the plate. In contrast, the effect of the radial offset of the plate is quite small.

## 1. Introduction

Due to their flexible and compact structure, spherical joints have attracted great interest in many engineering applications (e.g., industrial robots, machine tools and medical equipment [1,2]). However, the clearance between the ball and socket could have a detrimental effect on the accurate motion of multibody mechanical systems [3,4,5]. To achieve a high-precision kinematic performance of multibody systems, the micro-clearance in the spherical joints is required to be detected and used for compensating the motion error of the mechanical systems [6,7,8,9,10,11].

Many methods have been proposed to detect displacement or clearance over the last few decades (e.g., capacitive sensors [12], magnetic sensors [13,14,15], microwave sensors [16], optical sensors [17], and inductive sensors [18,19,20]). Compared with other methods, capacitive sensors have the advantages of high resolution and good dynamic performance [21,22,23,24,25]. Ahn [26] developed a cylindrical capacitive sensor (CCS) to measure both the axial and radial motion of a rotating spindle. The sensor includes eight arc segments with an arbitrary angular size. Anandan and George [27] proposed a wide-range capacitive sensor to detect both linear and angular displacements of a rotational shaft. The displacements are determined by measuring the capacitance of four cylindrical electrodes placed around the shaft. For spherical joints, Hu et al. [28] deployed a capacitive sensor to detect the clearance between the ball and the socket. The capacitive sensor consists of six point capacitors. A point capacitance model is proposed for calculating the capacitance values of the sensor. However, the theoretical capacitances have a relatively large deviation from the simulated counterparts. Wang et al. [29] proposed a spherical capacitive sensor to detect the clearance in spherical joints. The sensor includes eight spherical electrode plates and a ball. However, the fringe effect at the corners of the plates could produce capacitance errors and reduce the measurement accuracy. To reduce the fringe effect of the capacitive sensor, an improved spherical capacitive sensor was then developed [30]. The sensor has six spherical plates without the sharp corners and thus could greatly reduce the fringe effect. However, the measurement accuracy of the spherical capacitive sensor could be affected by the manufacturing and installation imperfectness of the sensor. Thus, further studies are required to analyze the measurement error produced by the manufacturing and installation.

This work presents a detail error analysis of a spherical capacitive sensor, mainly focusing on the capacitance errors caused by the shape of the ball and the capacitive plate, the axial and radial offset of the plate, and the inclined installation of the plate. The measurement principle of the spherical capacitive sensor is briefly introduced in the Section 2. Section 3 deduces mathematical models for the relation between capacitance errors and the shape of the ball and plates. Section 4 describes the mathematical models used to calculate the capacitance error caused by the plate installation. The simulation detail of the spherical capacitive sensor is given in Section 5. The theoretical and simulated capacitances of the sensor are presented and discussed in Section 6.

## 2. Measurement Principle of the Spherical Capacitive Sensor

In this section, we briefly review the improved spherical capacitive sensor for clearance measurement in spherical joints. More details can be found in [30]. The sensor is composed of six spherical capacitive plates (*CP_S_*_1_ ~ *CP_S_*_6_) and a ball (*CP_e_*), as shown in Figure 1. Six spherical capacitive plates are concentrically arranged in the spherical joint, and the spherical center of the plates is chosen as the origin *O* of the coordinate system *OXYZ*. The ball is used as the common electrode. Each spherical capacitive plate (*CP_S_*_1_ ~ *CP_S_*_6_) and the ball constitute a spherical capacitive sensor (*C*_1_ ~ *C*_6_). As such, three pairs of capacitive sensors are formed (i.e., *C*_1_ and *C*_3_ in the *X*-axis, *C*_2_ and *C*_4_ in the *Y*-axis, and *C*_5_ and *C*_6_ in the *Z*-axis). The differential capacitance of each pair (Δ*C_x_*, Δ*C_y_*, Δ*C_z_*) is used to detect the eccentric displacement of the ball along the corresponding directions (*δ_x_*, *δ_y_*, *δ_z_*)—that is, Δ*C_x_* of *C*_1_ and *C*_3_ for *δ_x_*, Δ*C_y_* of *C/*_2_ and *C*_4_ for *δ_y_*, Δ*C_z_* of *C*_5_ and *C*_6_ for *δ_z_*. The structural parameters of the spherical capacitive sensor are presented in Table 1.

In this work, two assumptions are made to simplify the mathematical model. First, the capacitor for the area element of the spherical capacitive sensor can be regarded as a parallel plate capacitor. Second, the fringe effect of the spherical capacitive sensor can be ignored. Therefore, the capacitance value of the spherical capacitive sensor can be calculated by
(1)$$C_{i}=\zeta \iint_{A_{i}}\frac{1}{h}dA\left(i=1,2\cdots 6\right)$$
where *ζ* represents the dielectric constant, *h* is the gap distance between the ball and the spherical capacitive plate, and *A* is the area of the spherical capacitive plate.

Substitute the structural parameters of the spherical capacitive sensors into Equation (1); the capacitance of six spherical capacitive sensors can be obtained by the following equations:(2)$$\left\{ \begin{array}{l} C_{i}=\frac{\zeta R^{2}}{d_{0}}F_{i}\left(\lambda_{x},\lambda_{y},\lambda_{z}\right),i=1,2\cdots 6\\ \lambda_{x}=\frac{\delta_{x}}{d_{0}},\lambda_{y}=\frac{\delta_{y}}{d_{0}},\lambda_{z}=\frac{\delta_{z}}{d_{0}} \end{array}\right.$$
where ζ is the dielectric constant of the medium between the ball and the spherical capacitive plates; *d*_0_ is the initial clearance between the spherical capacitive plate and the ball when the ball is at the origin *O* in the socket*; δ_x_*, *δ*_y_, and *δ_z_* represent the eccentric displacement of the ball along the X, Y, and Z axes, respectively; and *F_i_*(*λ_x_, λ_y_, λ_z_*) is the function related to the non-dimensional parameters (*λ_x_, λ_y_, λ_z_*).

By ignoring the higher-order items above the fifth order, the functional relationship between the differential capacitance value (Δ*C_x_*, Δ*C_y_*, Δ*C_z_*) and the eccentric displacement (*δ_x_*, *δ_y_*, *δ_z_*) can be obtained as follows:(3)$$\left\{ \begin{array}{l} \Delta C_{x}=C_{1}-C_{3}=\frac{\zeta R^{2}}{d_{0}}\Delta F_{x}\\ \Delta C_{y}=C_{2}-C_{4}=\frac{\zeta R^{2}}{d_{0}}\Delta F_{y}\\ \Delta C_{z}=C_{5}-C_{6}=\frac{\zeta R^{2}}{d_{0}}\Delta F_{z} \end{array}\right.$$
where
(4)$$\begin{array}{l} \Delta F_{x}\;\;=1.5708\lambda_{x}+1.3745\lambda_{x}{}^{3}+0.2945\lambda_{x}\left(\lambda_{y}{}^{2}+\lambda_{z}{}^{2}\right)+\ldots \\ \;\;\;\;\;\;\;\;\;\;\;\;\;\;\;1.2108\lambda_{x}{}^{5}+0.06136\lambda_{x}{\left(\lambda_{y}{}^{2}+\lambda_{z}{}^{2}\right)}^{2}+0.8181\lambda_{x}{}^{3}\left(\lambda_{y}{}^{2}+\lambda_{z}{}^{2}\right) \end{array}$$
(5)$$\begin{array}{l} \Delta F_{y}\;\;=1.5708\lambda_{y}+1.3745\lambda_{y}{}^{3}+0.2945\lambda_{y}\left(\lambda_{x}{}^{2}+\lambda_{z}{}^{2}\right)+\ldots \\ \;\;\;\;\;\;\;\;\;\;\;\;\;\;\;1.2108\lambda_{y}{}^{5}+0.06136\lambda_{y}{\left(\lambda_{x}{}^{2}+\lambda_{z}{}^{2}\right)}^{2}+0.8181\lambda_{y}{}^{3}\left(\lambda_{x}{}^{2}+\lambda_{z}{}^{2}\right) \end{array}$$
(6)$$\begin{array}{l} \Delta F_{z}\;\;=1.0744\lambda_{z}+0.4454\lambda_{z}{}^{3}+0.5426\lambda_{z}\left(\lambda_{x}{}^{2}+\lambda_{y}{}^{2}\right)+\ldots \\ \;\;\;\;\;\;\;\;\;\;\;\;\;\;0.1872\lambda_{z}{}^{5}+0.6956\lambda_{z}{\left(\lambda_{x}{}^{2}+\lambda_{y}{}^{2}\right)}^{2}+1.2908\lambda_{z}{}^{3}\left(\lambda_{x}{}^{2}+\lambda_{y}{}^{2}\right) \end{array}$$

After ignoring the non-linear errors introduced by the higher-order terms in Δ*C_x_*, Δ*C_y_*, and Δ*C_z_*, the designed spherical capacitive sensor can linearly measure the eccentric displacement of the ball in the spherical joint, and the corresponding equations can be expressed by
(7)$$\left\{ \begin{array}{l} \delta_{x}=\frac{d_{0}{}^{2}}{1.5708\zeta R^{2}}\Delta C_{x}\\ \delta_{y}=\frac{d_{0}{}^{2}}{1.5708\zeta R^{2}}\Delta C_{y}\\ \delta_{z}=\frac{d_{0}{}^{2}}{1.0744\zeta R^{2}}\Delta C_{z} \end{array}\right.$$

## 3. Measurement Error Caused by the Manufacture of the Capacitive Sensor

In the actual manufacturing process of spherical surfaces, spherical shape deviations occur due to the installation error and wear of the cutter. Once the shape deviation of a spherical electrode exists, there is a clearance deviation between the spherical capacitive plates and the ball, producing measurement errors. In this section, theoretical models are deduced to analyze the dependence of the measurement accuracy on the shape of the ball and the plates. Considering that the ellipsoid is the most common spherical shape, this work mainly focuses on measurement errors caused by the ellipsoidal shapes of the ball and the plates.

### 3.1. Effect of the Ball Shape

If the ball shape becomes an ellipsoid, the clearance between the spherical capacitive plate and the ball would vary with the motion of the ball in the socket, as shown in Figure 2. As a result, the functional relationship between the capacitance and the measured displacement in the spherical capacitive sensor is related to the ball attitude. Therefore, the shape deviation of the ball could produce attitude error, affecting the measurement accuracy of the spherical capacitive sensor.

To illustrate the effect of the ball shape on the clearance measurement, several parameters are defined in Figure 2a. The spherical center of the ball is fixed at the point *O’*. *r* is the radius of the ideal ball and *R* is the radius of the inner spherical surface of the spherical capacitive plate. *P* is a point on the surface of the ideal ball, and *P’* is the intersection point of the extension line of *O’P* and the ellipsoidal ball. *θ* is the angle between the *OP* and the *X* axis, *φ* is the angle between *OP*’s projection line and the Y axis in the YOZ plane. *Q* is the intersection of the extension line of *O’P’* and the spherical surface inside the spherical capacitive plate. *P’Q* is the clearance between the spherical capacitive plate and the ellipsoidal ball. *PP’* represents the shape deviation from the ideal ball at the point *P’*.

As shown in Figure 2b, the clearance between the ball and the spherical capacitive plate varies with the rotation of the ellipsoidal ball. *a* and *b* are half the length of the principal axes, corresponding to the semi-minor axis and the semi-major axis of the ellipsoid, respectively. The attitude angle *β* is the angle between the semi-major axis of the ellipsoid and the *X* axis. For simplification, we assume that the shortest radius of the ellipsoid equals to the radius of the ideal ball (i.e., *a* = *r*). Initially, the semi-major axis of the ellipsoid is along the X axis (i.e., *β* = 0°), and has the smallest clearance between the ball and the spherical capacitive plate. As the ellipsoid rotates to *β* = 90°, the clearance between the ellipsoid and spherical capacitive plates becomes largest. Therefore, the maximum attitude error can be obtained by analyzing the measurement errors at the ball attitudes of *β* = 0° and 90°.

The maximum radius deviation from the ideal ball is defined as follows:(8)Δr=b−a=b−r

The distance *PP’* of the ellipsoid at different attitude angles from the ideal ball can be obtained by
(9)$$\left\{ \begin{array}{l} L_{00}=\Delta r\cos^{2}\theta \\ L_{90}=\Delta r\sin^{2}\theta \end{array}\right.$$
where *L*_00_ is the distance *PP’* of the ellipsoid at the attitude angle of *β* = 0° and *L*_90_ is the distance *PP’* of the ellipsoid at *β* = 90°.

Next, the clearance *P’Q* between the spherical capacitive plate and the ellipsoidal ball with different attitudes is given by
(10)$$\left\{ \begin{array}{l} d_{00}=R-\delta_{y}\sin\theta \cos\varphi -\delta_{z}\sin\theta \sin\varphi -\delta_{x}\cos\theta -r-\Delta r\cos^{2}\theta \\ d_{90}=R-\delta_{y}\sin\theta \cos\varphi -\delta_{z}\sin\theta \sin\varphi -\delta_{x}\cos\theta -r-\Delta r\sin^{2}\theta \end{array}\right.$$
where *d*_00_ is the clearance *P’Q* of the ellipsoid at the attitude angle of *β* = 0° and *d*_90_ is the clearance *P’Q* of the ellipsoid at *β* = 90°.

After introducing the dimensionless coefficient *K*_00_ = *d*_00_/(*R − r*) = *d*_00_/*d*_0_ and *K*_90_ = *d*_90_/(*R − r*) = *d*_90_/*d*_0_, Equation (10) can be expressed as follows:(11)$$\left\{ \begin{array}{l} K_{00}=1-\lambda_{y}\sin\theta \cos\varphi -\lambda_{z}\sin\theta \sin\varphi -\lambda_{x}\cos\theta -\kappa_{\Delta r}\cos^{2}\theta \\ K_{90}=1-\lambda_{y}\sin\theta \cos\varphi -\lambda_{z}\sin\theta \sin\varphi -\lambda_{x}\cos\theta -\kappa_{\Delta r}\sin^{2}\theta \end{array}\right.$$
where *κ_Δr_* = Δ*r/(R − r)* = Δ*r/d*_0_, *λ_x_* = *δ_x_/d*_0_, and *λ_y_* = *δ_y_/d*_0_,*λ_z_* = *δ_z_/d*_0_.

Let *λ*_0_ = *λ_y_*sin*θ*cos*φ* + *λz*sin*θ*sin*φ* + *λ_x_*cos*θ* + *κ*_Δ_*_r_*cos^2^*θ*. Since λ_0_ < 1, 1/*K*_00_ can be expressed by
(12)$$\frac{1}{K_{00}}=\frac{1}{1-\lambda_{0}}=1+\lambda_{0}+\lambda_{0}{}^{2}+\lambda_{0}{}^{3}+\cdots$$

Substitute Equation (12) into Equation (1) and we can obtain the capacitance values of the spherical capacitive sensors *C_1_* and *C_3_* when the ellipsoid is at the attitude angle of *β* = 0°,
(13)$$C_{1}{}^{0}=\frac{\zeta }{d_{0}}\int_{0}^{\theta_{1}}\int_{0}^{2\pi }E\left(\theta ,\varphi \right)R^{2}\sin\theta d\theta d\varphi$$
(14)$$C_{3}{}^{0}=\frac{\zeta }{d_{0}}\int_{\pi -\theta_{1}}^{\pi }\int_{0}^{2\pi }E\left(\theta ,\varphi \right)R^{2}\sin\theta d\theta d\varphi$$
where $E(\theta ,\varphi )=\sum_{n=0}^{\infty }(\lambda_{x}\sin\theta \cos\varphi +\lambda_{y}\sin\theta \sin\varphi +\lambda_{z}\cos\theta +\kappa_{\Delta r}\cos^{2}\theta {)}^{n}$, and $\theta_{1}=1/6\pi$.

By neglecting higher-order terms beyond the fifth order, we can get the differential capacitance Δ*C_x_*^0^ for detecting the eccentric displacement along the X-axis when the ellipsoid is at the attitude angle of *β* = 0°:(15)$$\Delta C_{x}{}^{0}=\frac{\zeta R^{2}}{d_{0}}\Delta G_{x}{}^{0}\left(\lambda_{x},\lambda_{y},\lambda_{z}\right)$$
where
(16)$$\begin{array}{l} \Delta G_{x}{}^{0}=(1.5708+4.7922\kappa_{\Delta r}^{4}+4.2952\kappa_{\Delta r}^{3}+3.6325\kappa_{\Delta r}^{2}+2.7489\kappa_{\Delta r})\lambda_{x}\\ \;\;\;\;\;\;\;\;\;\;\;\;\;\;\;\;\;\;(3.6325\kappa_{\Delta r}^{2}+4.8433\kappa_{\Delta r}+1.3745)\lambda_{x}^{3}+\ldots \\ \;\;\;\;\;\;\;\;\;\;\;\;\;\;\;\;\;\;(2.0555\kappa_{\Delta r}^{2}+0.9817\kappa_{\Delta r}+0.2945)\lambda_{x}(\lambda_{y}^{2}+\lambda_{z}^{2})+\ldots \\ \;\;\;\;\;\;\;\;\;\;\;\;\;\;\;\;\;\;1.2108\lambda_{x}^{5}+0.0614\lambda_{x}(\lambda_{y}^{2}+\lambda_{z}^{2}){}^{2}+0.8181\lambda_{x}^{3}(\lambda_{y}^{2}+\lambda_{z}^{2}) \end{array}$$

Likewise, when the ellipsoid is at the attitude angle of *β* = 90°, the relation of differential capacitance and eccentric displacements can be obtained by
(17)$$\Delta C_{x}{}^{90}=\frac{\zeta R^{2}}{d_{0}}\Delta G_{x}{}^{90}\left(\lambda_{x},\lambda_{y},\lambda_{z}\right)$$
where
(18)$$\begin{array}{l} \Delta G_{x}{}^{90}=(1.5708+0.0614\kappa_{\Delta r}^{4}+0.0245\kappa_{\Delta r}^{3}+0.0982\kappa_{\Delta r}^{2}+0.3193\kappa_{\Delta r})\lambda_{x}\\ \;\;\;\;\;\;\;\;\;\;\;\;\;\;\;\;\;\;(0.2659\kappa_{\Delta r}^{2}+0.6545\kappa_{\Delta r}+1.3745)\lambda_{x}^{3}+\ldots \\ \;\;\;\;\;\;\;\;\;\;\;\;\;\;\;\;\;\;(0.092\kappa_{\Delta r}^{2}+0.1964\kappa_{\Delta r}+0.2945)\lambda_{x}(\lambda_{y}^{2}+\lambda_{z}^{2})+\ldots \\ \;\;\;\;\;\;\;\;\;\;\;\;\;\;\;\;\;\;1.2108\lambda_{x}^{5}+0.0614\lambda_{x}(\lambda_{y}^{2}+\lambda_{z}^{2}){}^{2}+0.8181\lambda_{x}^{3}(\lambda_{y}^{2}+\lambda_{z}^{2}) \end{array}$$

Therefore, the maximum attitude errors (*H_M_*) caused by shape deviation of the ball can be calculated by
(19)$$\left\{ \begin{array}{l} H_{M}=Max(\frac{\Delta G_{x}{}^{0}-\Delta G_{x}{}^{90}}{\Delta G_{x}{}^{90}})\\ \;\;0\leq \sqrt{\lambda_{x}^{2}+\lambda_{y}^{2}+\lambda_{z}^{2}}=\rho \end{array}\right.$$

### 3.2. Effect of the Plate Shape

If the inner surface of a spherical capacitive plate becomes ellipsoid, the clearance between the ball and plates are apparently different from that between the ideal spherical surfaces, producing the measurement error. For simplification, it is assumed that the inner surface shapes of *CP_S_*_1_ and *CP_S_*_3_ are the different parts of an ellipsoid, which has a semi-major radius of *a* and a semi-minor radius of *b*. As a result, the clearance between *CP_S_*_1_ and the ball is different from that between *CP_s_*_3_ and the ball. This difference becomes largest when the shape of *CP_S_*_1_ is the ellipsoid about the major axis and the shape of *CP_S_*_3_ is the ellipsoid about the minor axis, as shown in Figure 3.

As shown in Figure 3a, *Q*’ is the intersection of the extension line of *O*’*P* and the ellipsoidal plate, *QQ*’ is the deviation of the ellipsoid from the ideally spherical plate, and *PQ’* is the clearance between the ellipsoidal plate and the ball at the point *P*. The clearance can be obtained by
(20)$$\left\{ \begin{array}{l} d^{1}=R-\delta_{y}\sin\theta \cos\varphi -\delta_{z}\sin\theta \sin\varphi -\delta_{x}\cos\theta -r+\Delta r\cos^{2}\theta \\ d^{3}=R-\delta_{y}\sin\theta \cos\varphi -\delta_{z}\sin\theta \sin\varphi -\delta_{x}\cos\theta -r+\Delta r\sin^{2}\theta \end{array}\right.$$
where *d*^1^ is the clearance between *CP_S_*_1_ and the ball, *d*^3^ is the clearance between *CP_s_*_3_ and the ball, and Δ*r = b − a*.

Likewise, the functional relationship between the differential capacitance (Δ*C’_x_*) and the eccentric displacement of the ball can be obtained as follows:(21)$$\left\{ \begin{array}{l} \Delta C{{}^{\prime }}_{x}=\frac{\zeta R^{2}}{d_{0}}\left[a_{0}+a_{1}\lambda_{x}+C_{s}(\lambda_{y},\lambda_{z})+\Delta F_{s}(\lambda_{x},\lambda_{y},\lambda_{z})\right]\\ C_{s}=a_{2}{}^{\prime }(\lambda_{y}^{2}+\lambda_{z}^{2})+a_{4}{}^{\prime }{\left(\lambda_{y}^{2}+\lambda_{z}^{2}\right)}^{2}\\ \Delta F_{s}=a_{2}\lambda_{x}^{2}+a_{3}\lambda_{x}^{3}+a_{3}{}^{\prime }\lambda_{x}(\lambda_{y}^{2}+\lambda_{z}^{2})+\;a_{4}\lambda_{x}^{4}\;+a_{4}{}^{\prime \prime }\lambda_{x}^{2}(\lambda_{y}^{2}+\lambda_{z}^{2})+\cdots \\ \;\;\;\;\;\;\;\;\;\;\;\;\;\;\;\;\;\;1.2108\lambda_{x}^{5}\;\;+0.0614\lambda_{x}(\lambda_{y}^{2}+\lambda_{z}^{2}){}^{2}+0.8181\lambda_{x}^{3}(\lambda_{y}^{2}+\lambda_{z}^{2}) \end{array}\right.$$
where $a_{0}=-0.4537\kappa_{\Delta r}^{5}+0.5061\kappa_{\Delta r}^{4}-0.5662\kappa_{\Delta r}^{3}+0.6263\kappa_{\Delta r}^{2}-0.6263\kappa_{\Delta r}$
$a_{1}=2.3992\kappa_{\Delta r}^{4}-2.1598\kappa_{\Delta r}^{3}+1.8653\kappa_{\Delta r}^{2}-1.5708\kappa_{\Delta r}+1.5708$, $a_{2}=-0.5041\kappa_{\Delta r}^{3}+3.3295\kappa_{\Delta r}^{2}-1.6647\kappa_{\Delta r}$, $a_{2}{}^{\prime }=-0.3107\kappa_{\Delta r}^{3}+0.2142\kappa_{\Delta r}^{2}-0.1071\kappa_{\Delta r}$, $a_{3}=5.5019\kappa_{\Delta r}^{2}-2.7489\kappa_{\Delta r}+1.3745$, $a_{3}{}^{\prime }=1.0738\kappa_{\Delta r}^{2}-0.589\kappa_{\Delta r}+0.2945$, $a_{4}=-2.4742\kappa_{\Delta r}$, $a_{4}{}^{\prime }=-0.0212\kappa_{\Delta r}$, $a_{4}{}^{\prime \prime }=-0.9012\kappa_{\Delta r}$.

Compared with the ideally spherical plate (Equation (3)), the ellipsoidal plate (Equation (21)) produces three terms of additional capacitance errors (i.e., constant capacitance error, variable capacitance error, and higher-order capacitance error).

The constant value *a*_0_ in Equation (21) could bring an additional constant capacitance (Δ*C_s_*^0^) to the capacitance value of the sensor, leading to an additional constant capacitance error of the sensor. Δ*C_s_*^0^ can be calculated by
(22)$$\Delta C_{s}^{0}=\frac{\zeta R^{2}}{d_{0}}a_{0}$$

Since the polynomial *C_s_*(*λ_y_,λ_z_*) is related to the measured displacement (*λ_y_* and *λ_z_*) rather than *λ_x_* in Equation (21), an additional variable capacitance (Δ*C_s_)* is added to the differential capacitance (Δ*C_x_*) of the sensor. To explore the effect of the additional variable capacitance on the measurement accuracy, we introduce the maximum additional variable capacitance (Δ*C_s_^M^*), given by
(23)$$\left\{ \begin{array}{l} \Delta C_{s}^{M}=Max\left[\frac{\zeta R^{2}}{d_{0}}C_{s}(\lambda_{y},\lambda_{z})\right]\\ \;\;\;\;\;\;0\leq \sqrt{\lambda_{x}^{2}+\lambda_{y}^{2}+\lambda_{z}^{2}}=\rho \end{array}\right.$$

Due to the presence of a higher-order term Δ*F*_s_ related to the measured *λ_x_*, a higher-order capacitance error is added to the measured differential capacitance, increasing the nonlinear error of the capacitive sensor. The maximum capacitance error (*E_s_^M^*) is expressed by
(24)$$\left\{ \begin{array}{l} E_{s}^{M}=Max\left[\frac{\Delta F_{s}(\lambda_{x},\lambda_{y},\lambda_{z})}{a_{1}\lambda_{x}}\right]\\ \;\;\;\;\;\;0\leq \sqrt{\lambda_{x}^{2}+\lambda_{y}^{2}+\lambda_{z}^{2}}=\rho \end{array}\right.$$

## 4. Measurement Error Introduced by the Installation of the Capacitive Plates

During the assembly of the spherical plates and ball, the spherical centers of six capacitive plates may not overlap with each other, producing a measurement error of the capacitive sensor. The installation deviation of the capacitive plates could be classified into two types (i.e., offset installation deviation and angular installation deviation). The former is caused by the displacement of the capacitive plate along a certain direction while the latter comes from the angular inclination of the plate.

### 4.1. Effect of Offset Installation

Due to the axial symmetry of the spherical capacitive plate, the displacement of the plate can be divided into two orthogonal directions. One is along the symmetrical axis, the other is along the radial direction that is orthogonal to the symmetrical axis. Accordingly, the offset installation error can be divided into axial and radial offset installation errors, respectively.

#### 4.1.1. Axial Offset Installation

The axial displacement of the spherical capacitive plate is presented in Figure 4. The capacitive plate *CP_S_*_1_ moves a distance of *m* away from the ideal position along the positive x axis while the *CP_S_*_3_ is installed at the ideal position. Therefore, the electrode coordinate system *X’*_1_*Y’*_1_*Z’*_1_ in which *CP_S_*_1_ is located is shifted by *m* along the positive x axis. The mathematical conversion between the coordinate system *X’*_1_*Y’*_1_*Z’*_1_ and the coordinate system *OXYZ* can be given by
(25)$$\left[\begin{array}{l} x{{}^{\prime }}_{1}\\ y{{}^{\prime }}_{1}\\ z{{}^{\prime }}_{1}\\ 1 \end{array}\right]=\left[\begin{array}{cccc} 1 & 0 & 0 & m\\ 0 & 1 & 0 & 0\\ 0 & 0 & 1 & 0\\ 0 & 0 & 0 & 1 \end{array}\right]\left[\begin{array}{l} x\\ y\\ z\\ 1 \end{array}\right]$$

The capacitance of the capacitive sensor *CP_S_*_1_ with axial displacement can be obtained by substituting Equation (25) into Equation (2). Then, the differential capacitance value of *C*_x_ can be given by
(26)$$\Delta C_{x}^{m}=\frac{\zeta R^{2}}{d_{0}}[m_{0}+m_{1}\lambda_{x}+C_{m}(\lambda_{y},\lambda_{z})+\Delta F_{m}(\lambda_{x},\lambda_{y},\lambda_{z})]$$
(27)$$\begin{array}{l} \Delta F_{m}=m_{2}\lambda_{x}^{2}+m_{3}\lambda_{x}^{3}+m_{3}{}^{\prime }\lambda_{x}(\lambda_{y}^{2}+\lambda_{z}^{2})+\;m_{4}\lambda_{x}^{4}\;\;+m_{4}{}^{\prime \prime }\lambda_{x}^{2}(\lambda_{y}^{2}+\lambda_{z}^{2})+\cdots \\ \;\;\;\;\;\;\;\;\;\;\;\;\;\;\;\;\;\;\;\;\;\;\;\;\;\;\;\;\;\;\;1.2108\lambda_{x}^{5}\;\;+0.0614\lambda_{x}(\lambda_{y}^{2}+\lambda_{z}^{2}){}^{2}+0.8181\lambda_{x}^{3}(\lambda_{y}^{2}+\lambda_{z}^{2}) \end{array}$$
(28)$$C_{m}=m_{2}{}^{\prime }(\lambda_{y}^{2}+\lambda_{z}^{2})+m_{4}{}^{\prime }{\left(\lambda_{y}^{2}+\lambda_{z}^{2}\right)}^{2}\;$$
where $m_{0}=0.6054\kappa_{m}^{5}+0.6445\kappa_{m}^{4}+0.6872\kappa_{m}^{3}+0.734\kappa_{m}^{2}+0.7854\kappa_{m}$, $m_{1}=3.027\kappa_{m}^{4}+2.5779\kappa_{m}^{3}+2.0617\kappa_{m}^{2}+2.0617\kappa_{m}+1.5708$, $m_{2}=6.0541\kappa_{m}^{3}+2.5779\kappa_{m}^{2}+2.0617\kappa_{m}$, $m_{2}{}^{\prime }=0.4091\kappa_{m}^{3}+0.2687\kappa_{m}^{2}+0.1473\kappa_{m}$, $m_{3}=6.0541\kappa_{m}^{2}+2.5779\kappa_{m}+1.3745$, $m_{3}{}^{\prime }=1.2272\kappa_{m}^{2}+0.5374\kappa_{m}+0.2945$, $m_{4}=3.0271\kappa_{m}$, $m_{4}{}^{\prime }=0.0307\kappa_{m}$, $m_{4}{}^{\prime \prime }=1.2272\kappa_{m}$, $\kappa_{m}=m/(R-r)$.

Based on Equation (26), the axial offset installation of the spherical capacitance sensor will produce three terms of additional capacitance errors: the constant capacitance *EC_m_*^0^, the variable capacitance *VC_m_*, and higher-order capacitance *E_m_*. They are introduced by the constant term *m_0_*, the term *C_m_(λ_y_,λ_z_)*, and the higher-order term Δ*F_m_ (λ_x_, λ_y_, λ_z_)*, respectively.

The constant capacitance error *EC_m_^0^* can be obtained by
(29)$$EC_{m}^{0}=\frac{\zeta R^{2}}{d_{0}}m_{0}$$

To explore the effect of the variable capacitance *VC_m_* on the measurement accuracy, the maximum variable capacitance error *VC_m_^M^* is given by
(30)$$\left\{ \begin{array}{l} VC_{m}^{M}=Max\left[\frac{\zeta R^{2}}{d_{0}}C_{m}(\lambda_{y},\lambda_{z})\right]\\ \;\;\;\;\;\;0\leq \sqrt{\lambda_{x}^{2}+\lambda_{y}^{2}+\lambda_{z}^{2}}=\rho \end{array}\right.$$

To explore the effect of the higher-order capacitance *E_m_* on the measurement accuracy, the maximum higher-order capacitance error *E_m_^M^* is obtained by
(31)$$\left\{ \begin{array}{l} E_{m}^{M}=Max\left[\frac{\Delta F_{m}(\lambda_{x},\lambda_{y},\lambda_{z})}{m_{1}\lambda_{x}}\right]\\ \;\;\;\;\;\;0\leq \sqrt{\lambda_{x}^{2}+\lambda_{y}^{2}+\lambda_{z}^{2}}=\rho \end{array}\right.$$

#### 4.1.2. Radial Offset Installation

The radial displacement of a pair of spherical plates is described in Figure 5. Two typical cases of plate displacements are identified in this work: (1) the spherical capacitive plates *CP_S_*_1_ and *CP_S_*_3_ are misaligned, as shown in Figure 5a; (2) the spherical plates *CP_S_*_1_ and *CP_S_*_3_ are placed away from the ideal position along the same direction, as is presented in Figure 5b.

(A) Displacement of *CP_S_*_1_ and *CP_S_*_3_ along opposite directions

As shown in Figure 5a, the origin of the initial coordinate *XYZ* is located at the center of the ball. The positive *X*-axis is towards the center of the plate *CP_S_*_1_, while the positive *Y*-axis is perpendicular to the positive *X*-axis based on the right-hand rule. The spherical capacitive plate *CP_S_*_1_ is placed away from the ideal position with a distance of *h* along the positive Y direction, while *CP_S_*_3_ is along the negative Y direction. As a result, they are located in the coordinate systems *X*_1_*Y*_1_*Z*_1_ and *X*_3_*Y*_3_*Z*_3_, respectively. The distance between the centerline of *CP_S_*_1_ and that of *CP_S_*_2_ is 2*h*. Therefore, the mathematical relation between the offset coordinate systems *X*_1_*Y*_1_*Z*_1_ and *X*_3_*Y*_3_*Z*_3_ and the ideal coordinate system OXYZ can be obtained by
(32)$$\left[\begin{array}{l} x_{1}\\ y_{1}\\ z_{1}\\ 1 \end{array}\right]=\left[\begin{array}{cccc} 1 & 0 & 0 & 0\\ 0 & 1 & 0 & h\\ 0 & 0 & 1 & 0\\ 0 & 0 & 0 & 1 \end{array}\right]\left[\begin{array}{l} x\\ y\\ z\\ 1 \end{array}\right]\;\;\;\;\;\;\;\;\;\;\;\;\;\left[\begin{array}{l} x_{3}\\ y_{3}\\ z_{3}\\ 1 \end{array}\right]=\left[\begin{array}{cccc} 1 & 0 & 0 & 0\\ 0 & 1 & 0 & -h\\ 0 & 0 & 1 & 0\\ 0 & 0 & 0 & 1 \end{array}\right]\left[\begin{array}{l} x\\ y\\ z\\ 1 \end{array}\right]$$

The capacitance of capacitive sensor *CP_S_*_1_ and *CP_S_*_3_ can be obtained by substituting Equation (30) into Equation (2). When *CP_S_*_1_ and *CP_S_*_3_ are placed along the opposite radial directions, the differential capacitance of Δ*C^h^_x_* can be calculated by the following equation:(33)$$\Delta C_{x}^{h}=\frac{\zeta R^{2}}{d_{0}}\left[h_{1}\lambda_{x}+C_{h}(\lambda_{y},\lambda_{z})+\Delta F_{h}(\lambda_{x},\lambda_{y},\lambda_{z})\right]$$
(34)$$h_{1}=(0.0614\kappa_{h}^{4}+0.2945\kappa_{h}^{2}+1.5708)$$
(35)$$C_{h}=(0.0545\kappa_{h}^{3}+0.2154\kappa_{h})\lambda_{y}+(0.05453\kappa_{h})(\lambda_{y}\lambda_{z}^{2}+\lambda_{z}^{3})$$
(36)$$\begin{array}{l} \Delta F_{h}=\;(0.3682\kappa_{h}^{2}+0.2945)\lambda_{x}\lambda_{y}^{2}+(0.1227\kappa_{h}^{2}+0.2945)\lambda_{x}\lambda_{z}^{2}+\cdots \\ \;\;\;\;\;\;\;\;\;\;\;(1.0748\kappa_{h})\lambda_{y}\lambda_{x}^{2}+1.3745\lambda_{x}^{3}+\;\;1.2108\lambda_{x}^{5}+0.0614\lambda_{x}(\lambda_{y}^{2}+\lambda_{z}^{2}){}^{2}+0.8181\lambda_{x}^{3}(\lambda_{y}^{2}+\lambda_{z}^{2}) \end{array}$$
where *κ_h_* = *h/(R − r)*, *h_1_* is the coefficient of the first-order term *λ_x_*, *C_h_* (*λ_y_,λ_z_*) is a term related to the measured *λ_y_,* and*λ_z_*, Δ*F*_h_*(λ_x_,λ_y_,λ_z_)* is a higher-order term related to *λ_x_*, *λ_y_* and *λ_z_*.

According to Equation (31), two terms of additional capacitance errors are produced when *CP_S_*_1_ and *CP_S_*_3_ are misaligned (i.e., the variable capacitance *VC_h_^M^* and the higher-order capacitance *E_h_^M^* caused by the terms *C_h_(λ_y_,λ_z_*) and Δ*F_h_ (λ_x_, λ_y_, λ_z_*), respectively).

The maximum variable capacitance error (*VC_h_^M^*) can be given by
(37)$$\left\{ \begin{array}{l} VC_{h}^{M}=Max\left[\frac{\zeta R^{2}}{d_{0}}C_{h}(\lambda_{y},\lambda_{z})\right]\\ \;\;\;\;\;\;0\leq \sqrt{\lambda_{x}^{2}+\lambda_{y}^{2}+\lambda_{z}^{2}}=\rho \end{array}\right.$$

The maximum higher-order capacitance error (*E_h_^M^*) can be expressed by
(38)$$\left\{ \begin{array}{l} E_{h}^{M}=Max\left[\frac{\Delta F_{m}(\lambda_{x},\lambda_{y},\lambda_{z})}{h_{1}\lambda_{x}}\right]\\ \;\;\;\;\;\;0\leq \sqrt{\lambda_{x}^{2}+\lambda_{y}^{2}+\lambda_{z}^{2}}=\rho \end{array}\right.$$

(B) Displacement of *CP_S_*_1_ and *CP_S_*_3_ along the same direction

As presented in Figure 5b, both *CP_S_*_1_ and *CP_S_*_3_ are placed away from ideal position with a distance of *n* along the positive Y direction, and located in new coordinate system *X’Y’Z’*. Therefore, the mathematical relation between the offset coordinate system *X’Y’Z’* and the ideal coordinate system *OXYZ* can be expressed by
(39)$$\left[\begin{array}{l} x{}^{\prime }\\ y{}^{\prime }\\ z{}^{\prime }\\ 1 \end{array}\right]=\left[\begin{array}{cccc} 1 & 0 & 0 & 0\\ 0 & 1 & 0 & n\\ 0 & 0 & 1 & 0\\ 0 & 0 & 0 & 1 \end{array}\right]\left[\begin{array}{l} x\\ y\\ z\\ 1 \end{array}\right]$$

Next, the capacitance of capacitive sensor *CP_S_*_1_ and *CP_S_*_3_ can be obtained by substituting Equation (34) into Equation (2). As such, the differential capacitance of Δ*C^n^_x_* can be calculated by the following equation when *CP_S_*_1_ and *CP_S_*_3_ are placed along the same radial direction:(40)$$\Delta C_{x}^{n}=\frac{\zeta R^{2}}{d_{0}}\left[n_{1}\lambda_{x}+\Delta F_{n}(\lambda_{x},\lambda_{y},\lambda_{z})\right]$$
(41)$$n_{1}=1.5708+0.06136\kappa_{n}^{4}+0.2945\kappa_{n}^{2}$$
(42)$$\begin{array}{l} \Delta F_{n}=(0.2454\kappa_{n}^{3}+0.589\kappa_{n})\lambda_{x}\lambda_{y}+0.2954\lambda_{x}(\lambda_{y}^{2}+\lambda_{z}^{2})+(0.1227\kappa_{n}^{2})\lambda_{x}(3\lambda_{y}^{2}+\lambda_{z}^{2})+\cdots \\ \;\;\;\;\;\;\;\;\;\;\;(0.8181\kappa_{n}^{2}+1.3744)\lambda_{x}^{3}+(0.2454\kappa_{n})\lambda_{x}\lambda_{y}(6.668\lambda_{x}^{2}+\lambda_{y}^{2}+\lambda_{z}^{2})+\cdots \\ \;\;\;\;\;\;\;\;\;\;\;1.2108\lambda_{x}^{5}+0.0614\lambda_{x}(\lambda_{y}^{2}+\lambda_{x}^{2}){}^{2}+0.8181\lambda_{x}^{3}(\lambda_{y}^{2}+\lambda_{x}^{2}) \end{array}$$
where *κ_n_*=*n/(R-r)*, *n_1_* is the coefficient of the first-order term *λ_x_*, and Δ*F*_n_*(λ_x_,λ_y_,λ_z_)* is a higher-order term related to *λ_x_*, *λ_y_* and *λ_z_*.

It can be seen from Equation (35) that an additional higher-order capacitance is produced when *CP_S_*_1_ and *CP_S_*_3_ are offset along the same radial direction. The maximum higher-order capacitance error (*E_n_^M^*) can be expressed by
(43)$$\left\{ \begin{array}{l} E_{n}^{M}=Max\left[\frac{\Delta F_{n}(\lambda_{x},\lambda_{y},\lambda_{z})}{n_{1}\lambda_{x}}\right]\\ \;\;\;\;\;0\leq \sqrt{\lambda_{x}^{2}+\lambda_{y}^{2}+\lambda_{z}^{2}}=\rho \end{array}\right.$$

### 4.2. Effect of Inclined Installation

The inclined installation of the capacitive plate is shown in Figure 6. The spherical capacitive plate *CP_S_*_1_ has an inclined angle of *α* from the ideal position and thus is located in a new coordinate system *X”*_1_*Y”*_1_*Z”*_1_. The mathematical relation between the coordinate system *X”*_1_*Y”*_1_*Z”*_1_ and the ideal coordinate system OXYZ can be obtained by
(44)$$\left(\begin{array}{l} x{{}^{\prime \prime }}_{1}\\ y{{}^{\prime \prime }}_{1}\\ z{{}^{\prime \prime }}_{1}\\ 1 \end{array}\right)=\left(\begin{array}{cccc} \cos\alpha & -\sin\alpha & 0 & R\cos\alpha -R\\ \sin\alpha & \cos\alpha & 0 & R\sin\alpha \\ 0 & 0 & 1 & 0\\ 0 & 0 & 0 & 1 \end{array}\right)\left(\begin{array}{l} x\\ y\\ z\\ 1 \end{array}\right)$$

The capacitance of capacitive sensor *CP_S_*_1_ and *CP_S_*_3_ can be obtained by substituting Equation (44) into Equation (2). Thus, when *CP_S_*_1_ has an inclined angle from the ideal position, the differential capacitance of Δ*C^h^_x_* can be calculated by
(45)$$\Delta C_{x}^{p}=\frac{\zeta R^{2}}{d_{0}}\Delta F_{n}(\lambda_{x},\lambda_{y},\lambda_{z})=\frac{\zeta R^{2}}{d_{0}}\left[p_{0}+p_{1}\lambda_{x}+C_{P}(\lambda_{y},\lambda_{z})+\Delta F_{p}(\lambda_{x},\lambda_{y},\lambda_{z})\right]$$
where $p_{0}=-0.01875+0.01875cos\alpha +0.000427cos^{2}\alpha +0.000031sin^{2}\alpha +0.00001cos^{3}\alpha +0.000002cos\alpha sin^{2}\alpha$; $p_{1}=0.002509sin^{2}\alpha +0.034125cos^{2}\alpha +0.001289cos^{3}\alpha +0.749984cos\alpha +0.000276cos\alpha sin^{2}\alpha$
$$\begin{array}{ll} C_{P}(\lambda_{y},\lambda_{z}) & =(0.7854-0.000092sin^{3}\alpha -0.749984sin\alpha -0.031616cos\alpha sin\alpha -0.001104cos^{2}\alpha sin\alpha )\lambda_{y}\\ & +\;(-0.734046+0.050189cos^{2}\alpha +0.003682cos^{3}\alpha +0.682504sin^{2}\alpha +0.044179cos\alpha sin^{2}\alpha )\lambda_{y}{}^{2}\\ & +\;(0.687223-0.687223sin^{3}\alpha -0.147262cos^{2}\alpha sin\alpha )\lambda_{y}{}^{3}+(-0.003682+0.003682cos\alpha )\lambda_{z}{}^{2}\\ & +\;(0.147262-0.147262sin\alpha )\lambda_{y}\lambda_{z}{}^{2} \end{array}$$
$$\begin{array}{ll} \Delta F_{p}(\lambda_{x},\lambda_{y},\lambda_{z}) & =(-0.053871+0.682504cos^{2}\alpha +0.051542cos^{3}\alpha +0.050189sin^{2}\alpha +0.011044cosasin^{2}\alpha )\lambda_{x}{}^{2}\\ & +(0.687223cos^{3}\alpha +0.147262cos\alpha sin^{2}\alpha )\lambda_{x}{}^{3}\;\;\;\;\;\;\;\;\;\;\;\;\;\;\;\;\;\;\;\;\;\;\;\;\;\;\;\;\;\;\;\;\;\;\;\;\;\;\;\;\;\;\;\;\;\;\;\;\;\\ & +(-0.088357cos^{2}\alpha sin\alpha -1.26463\cos\alpha sin\alpha -0.00736sin^{3}\alpha )\lambda_{y}\lambda_{x}\\ & +\;\;(0.147262-0.147262sin^{3}\alpha -1.76714cos^{2}\alpha sin\alpha )\lambda_{y}\lambda_{x}{}^{2}+(0.147262cos\alpha )\lambda_{z}{}^{2}\lambda_{x}\;\\ & \;+\;(0.147262cos^{3}\alpha +1.76714cos\alpha sin^{2}\alpha )\lambda_{y}{}^{2}\lambda_{x} \end{array}$$

Therefore, the inclined installation of the plate *CP_S_*_1_ could produce three terms of additional capacitance errors: the constant capacitance error indicated by the constant term *p_0_*, the variable capacitance error represented by the term *C_p_(λ_y_,λ_z_)*, and the higher-order capacitance error indicated by higher-order term Δ*F_p_(λ_x_,λ_y_,λ_z_)*.

The constant error capacitance *EC_p_*^0^ can be expressed by
(46)$$EC_{p}^{0}=\frac{\zeta R^{2}}{d_{0}}p_{0}$$

The maximum variable capacitance (*VC_P_^M^*) can be given by
(47)$$\left\{ \begin{array}{l} VC_{p}^{M}=Max\left[\frac{\zeta R^{2}}{d_{0}}C_{p}(\lambda_{y},\lambda_{z})\right]\\ \;\;\;\;\;0\leq \sqrt{\lambda_{x}^{2}+\lambda_{y}^{2}+\lambda_{z}^{2}}=\rho \end{array}\right.$$

The maximum higher-order capacitance error (*E_p_^M^*) can be calculated by
(48)$$\left\{ \begin{array}{l} E_{p}^{M}=Max\left[\frac{\Delta F_{p}(\lambda_{x},\lambda_{y},\lambda_{z})}{p_{1}\lambda_{x}}\right]\\ \;\;\;\;\;0\leq \sqrt{\lambda_{x}^{2}+\lambda_{y}^{2}+\lambda_{z}^{2}}=\rho \end{array}\right.$$

## 5. Simulation Setup

In order to explore the effect of the shape and installation error of a spherical capacitive sensor on the clearance measurement, the software Ansoft Maxwell was used to simulate the capacitance of the sensor with installation and shape errors. For simplification, only spherical capacitive plates *CP_S_*_1_ and *CP_S_*_3_ were examined, forming a pair of capacitive sensors *C*_1_ and *C*_3_ in the *X*-axis. The differential capacitance Δ*C_x_* of *C*_1_ and *C*_3_ was calculated at various eccentricities of the ball. The simulation model of the capacitive sensor is presented in Figure 7. The ideal radius of the ball was 24.8 mm and the ideal inner radius of the plates was 25 mm. Two plates had a central angle of 0°~30° and a thickness of 0.5 mm. The material of the sensor was set to be brass and the medium between the ball and the plates was assumed to be air. The eccentric displacement of the ball varied from −40 to 40 μm along the *X* axis. Thus, the corresponding eccentricity was within the range of −0.2 ~ 0.2.

## 6. Results and Discussions

### 6.1. Effect of the Manufacture Deviation of Spherical Capacitive Sensor

#### 6.1.1. Measurement Error Caused by the Ball Shape

The capacitance of the sensor would vary with the rotation of the ball if the shape of the ball becomes ellipsoid. This could lead to an attitude error of the ball. Figure 8 shows the theoretical relation of the maximum attitude error and the eccentricity of the ellipsoidal ball. The ellipsoid has a shape of Δ*r* = 1, 2, and 5 μm, respectively. Hereinafter, the maximum value of the theoretical capacitance error is obtained from 1000 × 1000 points on a sphere for each eccentricity of the ball, which is generated using the sphere function in Matlab. Although the maximum attitude error *H_m_* increases with the rising eccentricity *ρ* of the ball, the increase of *H_m_* in three cases is very slight, within the range of 0 ≤ *ρ* ≤ 0.2. This is due to the differential arrangement of the capacitive plates. The differential capacitance of the sensor has a good linear relationship with the eccentric displacement of the ball. At Δ*r* = 1 μm, the maximum attitude error caused by the shape deviation of the ball is less than 0.8%. With an increase of Δ*r* to 2 μm, the maximum attitude error is between approximately 1.52% and 1.61%. Further, at Δ*r* = 5 μm, the maximum attitude error reaches 4.0%. Therefore, it is suggested that the shape deviation of the ball be within 1 μm to improve the measurement accuracy of the spherical capacitive sensor.

Figure 9 presents the dependence of the differential capacitance Δ*C_x_* on the eccentricity of the ellipsoidal ball at attitudes of *β* = 0° and 90°. The ellipsoidal ball has a shape deviation of Δ*r* = 10 μm and the eccentric displacement of the ball is from -40 to 40 μm along the X axis. The simulation results exhibit a good agreement with the theoretical counterparts, suggesting the feasibility of the proposed mathematical model for the ellipsoidal ball. The differential capacitances at *β* = 0° and 90° (Δ*C_x_*^0^ and Δ*C_x_*^90^) rise with the increasing eccentricity of the ellipsoidal ball. When the ball is located at the origin (*δ_x_* = 0), the differential capacitance at *β* = 0° is equal to that at 90°. However, when the ball has an eccentric displacement, there is a deviation between the differential capacitance at *β* = 90° and that at *β* = 0°. The difference becomes larger with the increasing eccentricity of the ellipsoidal ball, reaching at 7.7% at an eccentric displacement of *δ_x_* = 40 μm.

#### 6.1.2. Measurement Error Produced by the Plate Shape

The ellipsoidal plate was able to produce a constant capacitance error, variable capacitance error, and higher-order capacitance error to the measured differential capacitances of the spherical capacitive sensor. Figure 10 shows the theoretical capacitance error when the shape of the capacitive plates becomes ellipsoid. Several observations can be made. First, the maximum higher-order capacitance error increases slightly with the rising shape error of the plate Δ*r* from 0 to 6 μm (Figure 10a). At Δ*r* = 6 μm, the maximum higher-order capacitance error exhibits an exponential increase to about 4.2%, as the eccentricity of the ball goes up from 0 to 0.2. Compared with the ideal plate (Δr = 0), the increase of the maximum higher-order capacitance error for the ellipsoidal plate at Δ*r* = 2 μm is less than 0.2%. Therefore, the increased higher-order capacitance error can be ignored when the shape deviation of the plate Δr is no more than 2 μm. Second, the variable capacitance error rises significantly with the increasing Δ*r* (Figure 10b). At Δ*r* = 1 μm, the variable capacitance error is less than 0.7 × 10^−4^ pF when the eccentricity of the ball is in the range of 0 ≤ *ρ* ≤ 0.2. At Δ*r* = 6 μm, the variable capacitance error rises to about 3.4 × 10^−3^ pF at *ρ* = 0.2. Third, the constant capacitance error exhibits an approximately linear relationship with the shape deviation of the capacitive plate (Figure 10c). Compared with the variable capacitance error, the absolute value of the constant capacitance error is much larger, reaching about 0.1 pF at Δ*r* = 1 μm. However, it should be pointed out that the constant capacitance error can be eliminated by the calibration of the capacitive sensor.

Figure 11 shows the dependence of the differential capacitance Δ*C_x_* on the eccentric displacement of the ball when the capacitive plates become ellipsoid. The eccentric displacement varies from −40 to 40 μm along the X axis. It can be seen that the theoretical results exhibit a good agreement with the simulated counterparts, suggesting the feasibility of the proposed mathematical model for the ellipsoidal capacitive plates. Moreover, the differential capacitance of the ellipsoidal plates (Δ*r* = 10 μm) is smaller than that of the ideal plates (Δ*r* = 0). This can be mainly ascribed to the negative constant capacitance error mentioned above. The difference between the capacitance value of the ellipsoidal plates and that of the ideal plates exhibits an evident increase with the rising eccentricity of the ball.

### 6.2. Effect of the Installation Deviation of Spherical Capacitive Plates

#### 6.2.1. Measurement Error Caused by the Offset Installation

Once one capacitive plate *CP_S_*_1_ has an axial offset from the ideal position, it could introduce three kinds of capacitance errors to the differential capacitance of the spherical capacitive sensor *C_x_*. Figure 12 presents theoretical capacitance errors of the capacitive sensor *C_x_* at different offsets of *CP_S_*_1_. The displacement *m* of *CP_S_*_1_ increases from 0 to 10 µm along the *X* axis. The following observations can be drawn. First, the maximum higher-order capacitance error exhibits an increase with the rising offset *m* from 0 to 10 µm (Figure 12a). At *m* = 2 μm, the maximum higher-order capacitance error is less than 4% when the eccentricity of the ball is within the range of 0 ≤ *ρ* ≤ 0.2. And compared with the ideal condition (*m* = 0 μm), the increase of the maximum higher-order capacitance error for *m* = 2 μm is less than 0.4%. Second, the variable capacitance error exhibits a significant growth with the increase of the axial offset of the capacitive plate (Figure 12b), especially at a large eccentricity of the ball. At the axial offset of *m* = 2 μm, the variable capacitance error is less than 1.8 × 10^−3^ pF. Since the sensitivity of the spherical capacitive sensor is of 0.22 pF/μm [22], the corresponding displacement variation is less than 10 nm. Third, the constant capacitance error exhibits a linear relationship with the axial offset of the capacitive plate. Although the constant capacitance error could reach 0.1 pF at *m* = 1 μm, it can be eliminated by simple calibration of the capacitive sensor.

Figure 13 compares the theoretical and simulated values of the differential capacitance of Δ*C_x_^m^* at various axial offsets *m* of the capacitive plate *CP_S_*_1_. The axial displacements examined are 0, 10, and 20 μm, respectively. The eccentric displacement of the ball varies from −40 to 40 μm along the X axis. It can be observed that the theoretical results agree well with the simulated counterparts, suggesting the feasibility of the proposed theoretical model. Moreover, the differential capacitance Δ*C_x_^m^* for the axial offset of the plate (*m* = 10 and 20 μm) is smaller than that of the ideal position of the plate (*m* = 0). This can be mainly ascribed to the negative constant capacitance error mentioned above.

The radial offset of the capacitive plate can be divided into two regimes: the misaligned displacement of a pair of plates, and the identical displacement of the two plates. Figure 14 shows the capacitance error of the capacitive sensor *C_x_* at various eccentricities of the ball when capacitive plates *CP_S_*_1_ and *CP_S_*_3_ are radial offset from the ideal position and misaligned with each other. *CP_S_*_1_ has a radial displacement of 2~10 μm along the positive Y axis while *CP_S_*_3_ is placed with the same distance along the negative Y axis. Two kinds of additional capacitance errors are produced (i.e., higher-order capacitance error and variable capacitance error). As the radial displacement *h* of the capacitive plate increases from 0 to 10 μm, the values of the higher-order capacitance error all collapse into one line. This indicates that the higher-order capacitance error is independent on the radial displacement of the plates. In contrast, the variable capacitance error exhibits a remarkable growth with the increase of the radial displacement *h* of the plates. At *h* = 4 μm, the variable capacitance error is less than 1.2 × 10^−2^ pF, and the displacement error is less than 55 nm. Further increase *h* to 10 μm and the variable capacitance error becomes 6.0 × 10^−2^ pF at the eccentricity of the ball *ρ* = 0.2, five times of that of *h* = 4 μm. It should be pointed out that the variable capacitance error for radial offset of the plates is much larger than that for the axial offset of the plates.

Figure 15 compares the theoretical and simulated differential capacitance of the capacitive sensor (*C_x_*) at various eccentric displacements of the ball when the capacitive plates *CP_S_*_1_ and *CP_S_*_3_ are radial offset and misaligned with each other. *CP_S_*_1_ is offset 50 and 100 μm along the positive Y axis, while *CP_S_*_3_ is offset with the same distance along the negative *Y* axis. The eccentric displacement of the ball varies from −40 to 40 μm along the X axis. It can be seen that the theoretical results agree well with the simulated counterparts, suggesting the feasibility of the proposed mathematical model for the misaligned radial offset of the plates. Moreover, the differential capacitance exhibits a slight variation with the increase of the radial offset *h* of the plates. Although the variable capacitance error mentioned above grows significantly with the increase of the radial offset, the total value is still quite small.

Figure 16 shows the theoretical capacitance error when both *CP_S_*_1_ and *CP_S_*_3_ are radial offset along the same direction. The radial displacement *n* of two plates varies from 0 to 100 μm along the positive Y axis. The eccentricity of the ball varies from 0 to 0.2. According to Equation (35), the higher-order capacitance error is produced in this case. It can be seen that the maximum higher-order capacitance error exhibits a slight increase with the growth of the radial offset *n* of two capacitive plates. Even at *n* = 100 μm, the increase of the higher-order capacitance error is less than 0.3%, in comparison with the ideal position of the plates. This suggests that the identical radial offset of two plates has a neglected effect on the measurement accuracy of the sensor.

Figure 17 presents the theoretical and simulated differential capacitance of the capacitive sensor *C_x_* at various eccentric displacements of the ball when both *CP_S_*_1_ and *CP_S_*_3_ are radial offset along the Y axis. The radial displacement *n* of the plate is set to 50 and 100 μm, respectively. The eccentric displacement of the ball varies from -40 to 40 μm along the X axis. It can be observed that the theoretical results exhibit a good agreement with the simulated counterparts, indicating the feasibility of the proposed mathematical model for radial offsetting two plates along the same direction. Moreover, the differential capacitance varies slightly with the increase of the radial offset *n* of the plates. At *n =* 100 μm, a slight increase of the differential capacitance can be observed only at the eccentric displacement of the ball δ*_x_* ≥ 30 μm, compared with that for the ideal position of the plates.

#### 6.2.2. Measurement Error Produced by the Angular Installation

Once the capacitive plate is inclined with an angle to the ideal position, three kinds of capacitance errors can be produced, including higher-order capacitance errors, variable capacitance errors, and constant capacitance errors. Figure 18 shows the theoretical capacitance error of *C_x_* when the capacitive plate *CP_S_*_1_ is tilted. The inclined angle α of *CP_S_*_1_ increases from 0° to 0.4° and the eccentricity of the ball varies from −0.2 to 0.2. Three observations can be made. First, the higher-order capacitance error presents a slight increase with the growth of the inclined angle (*α*) of the plate *CP_S_*_1_. At an inclined angle of *α* = 0.1°, the increased higher-order capacitance error is less than 0.3% compared with the ideal counterpart. Second, the variable capacitance error rises significantly with the increase of the inclined angle. At the inclined angle of *α* = 0.02°, the variable capacitance error has an approximately linear relationship with the eccentricity of the ball *ρ*, reaching −2.4 × 10^−2^ pF at *ρ* = 0.2. Further, the corresponding displacement error is less than 0.11 μm. By increasing *α* to 0.1°, the variable capacitance error could reach −0.12 pF at *ρ* = 0.2, five times that of *α* = 0.02°. Third, the constant capacitance error exhibits an exponential growth with the increase of the inclined angle of the plate. The value of the constant capacitance error is of 0.1 pF at *α* = 0.12° and could reach to 0.95 pF at *α* = 0.36°.

Figure 19 compares the theoretical and simulated values of differential capacitance Δ*C_x_^P^* at various eccentric displacements of the ball when the plate *CP_S_*_1_ is inclined with an angle to the ideal position. The inclined angles are 0.2° and 0.4°, respectively. The eccentric displacement of the ball varies from −40 to 40 μm. It can be seen that the theoretical values agree well with the simulated counterparts, indicating the feasibility of the proposed mathematical model for the inclined installation of the capacitive plates. Moreover, the differential capacitance Δ*C_x_^P^* exhibits a slight growth as the inclined angle *α* rises from 0° to 0.2°. By further increasing *α* to 0.4, a remarkable growth of Δ*C_x_^P^* can be observed. This can be mainly ascribed to the exponential growth of the constant capacitance error mentioned above.

To obtain a good measurement accuracy of the spherical capacitive sensor, the manufacturing and installation imperfectness should be maintained in a small certain range. The spherical shape deviations of both plates and ball are suggested to be kept within 1 μm. The axial offset of the capacitive plate is required to be within 2 μm. The radial offset of the plate has limited effect on the measurement accuracy of the capacitive sensor due to the differential structure of the sensor. The effect of the inclined angle of the plate is quite significant, which is required to be controlled within 0.2°.

## 7. Conclusions

This work mainly explores the effects of manufacturing and installation imperfectness on the measurement accuracy of a spherical capacitive sensor. The detail mathematical models are built for calculating and analyzing the capacitance errors of the capacitive sensor. A simulation with the software Ansoft Maxwell was conducted to validate the feasibility of the proposed mathematical models. The main conclusions are given as follows.

(1)If the shape of the ball becomes ellipsoid, the capacitance of the spherical capacitive sensor could change with the rotation of the ball in a spherical joint, which leads to a large attitude error. If the shape error of the ball reaches 10 μm, the attitude error is about 7.7% at the eccentric displacement of ball *δ_x_* = 40 μm.(2)The ellipsoidal plate can produce higher-order capacitance errors, variable capacitance errors, and constant capacitance errors. For the shape error of the plate Δ*r* = 10 μm, the total capacitance error is of 15% at the eccentric displacement of ball *δ_x_* = 40 μm. This mainly comes from the negative constant additional capacitance.(3)The axial offset of the plate can generate higher-order capacitance errors, variable capacitance errors, and constant capacitance errors. Compared with the ideal position, the differential capacitance for the axial offset of the plate *m* = 10 μm is reduced by 17.2% at the eccentric displacement of ball *δ_x_* = 40 μm. This reduction can be mainly ascribed to the negative constant additional capacitance.(4)If a pair of plates is radial offset and misaligned, a higher-order capacitance error and variable capacitance error are produced. For the radial offset of the plate *h* = 100 μm, the total capacitance error is of 5.2% at the eccentric displacement of ball *δ_x_* = 40 μm.(5)If a pair of plates is radial offset and aligned, only a higher-order capacitance error is produced. At the eccentric displacement of ball *δ_x_* = 40 μm, the differential capacitance for the radial offset of *n* = 50 μm is increased by 1.3% compared with that for the ideal position.(6)The inclined capacitive plate could produce a higher-order capacitance error and variable capacitance error. For the inclined angle of the plate *α* = 0.2°, the total capacitance error is about 5% at the eccentric displacement of ball *δ_x_* = 40 μm.

## Figures and Tables

**Figure 1 micromachines-11-00837-f001:**
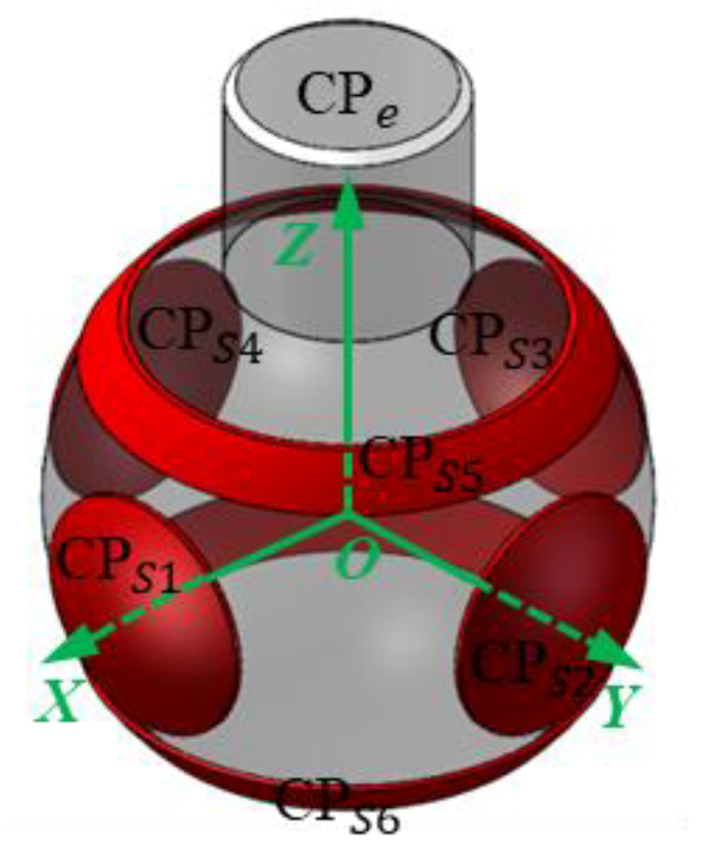
Geometry of the spherical capacitive sensor.

**Figure 2 micromachines-11-00837-f002:**
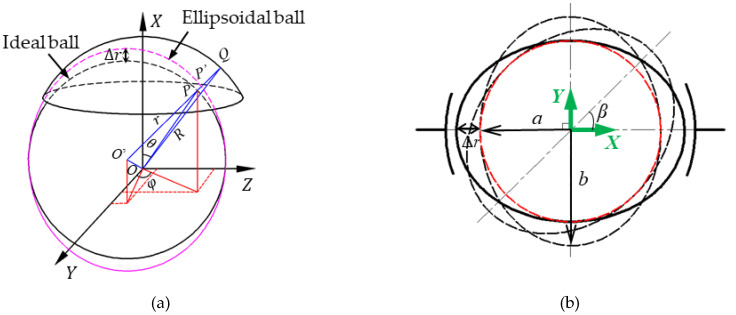
Sketch of the clearance between the ellipsoidal ball and the spherical plate. (**a**) 3D sketch, (**b**) 2D projection in the *XOY* plane as the ball rotates at various attitudes.

**Figure 3 micromachines-11-00837-f003:**
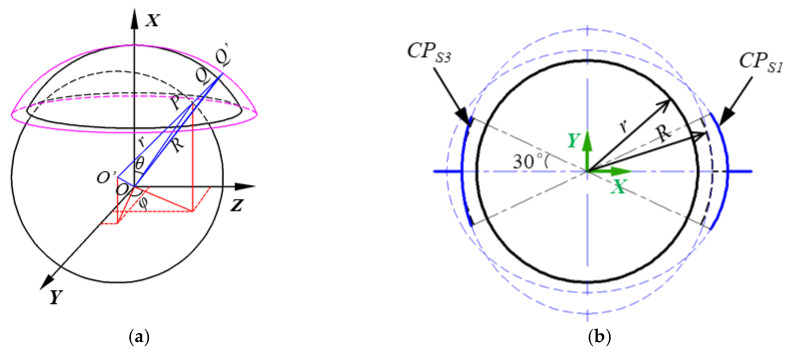
Sketch of the clearance between the spherical ball and the ellipsoidal plates. (**a**) 3D sketch, (**b**) 2D projection in the *XOY* plane.

**Figure 4 micromachines-11-00837-f004:**
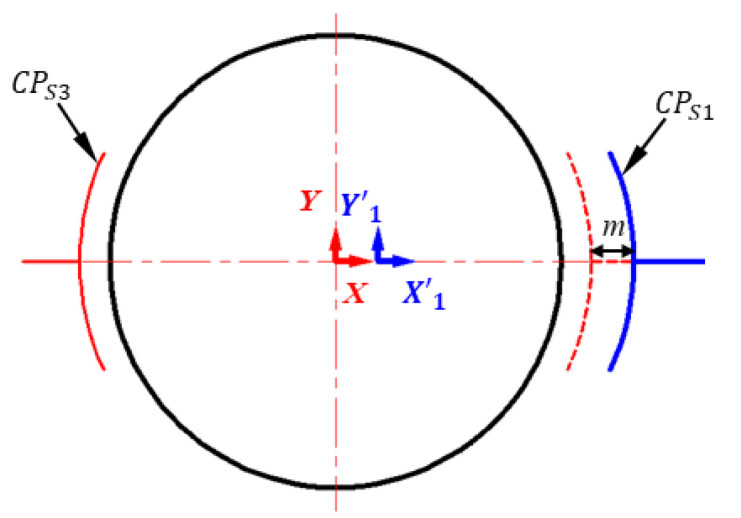
2D sketch of the spherical capacitance sensor *C_x_* when the spherical capacitive plate *CP_S_*_1_ is axial offset from the ideal position.

**Figure 5 micromachines-11-00837-f005:**
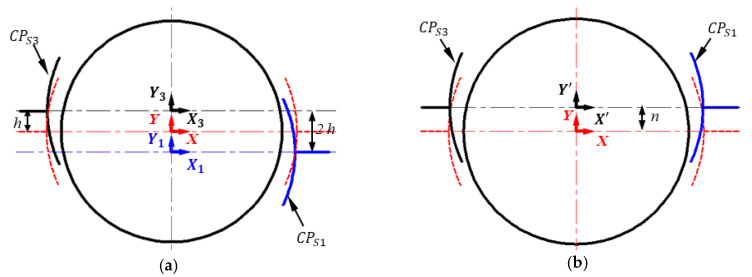
2D sketch of the spherical capacitance sensor *C_x_* when the spherical capacitive plate *CP_S_*_1_ is radial offset from the ideal position. (**a**) *CP_S_*_1_ and *CP_S_*_3_ are misaligned, (**b**) *CP_S_*_1_ and *CP_S_*_3_ are aligned.

**Figure 6 micromachines-11-00837-f006:**
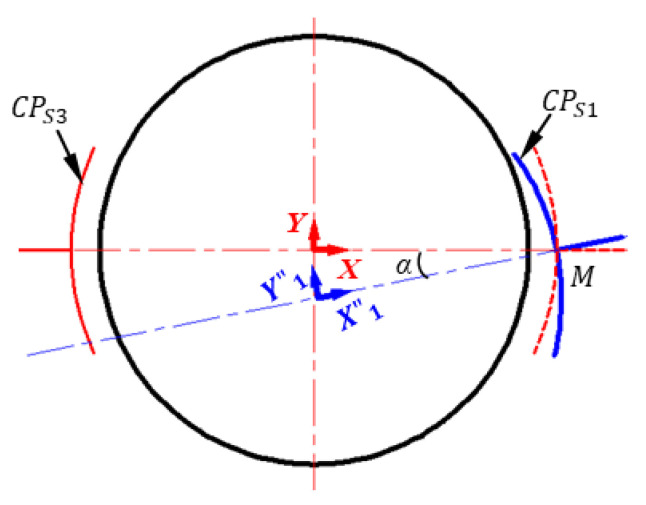
2D sketch of spherical capacitive sensor *C_x_* with the inclined installation of the plate *CP_S_*_1_.

**Figure 7 micromachines-11-00837-f007:**
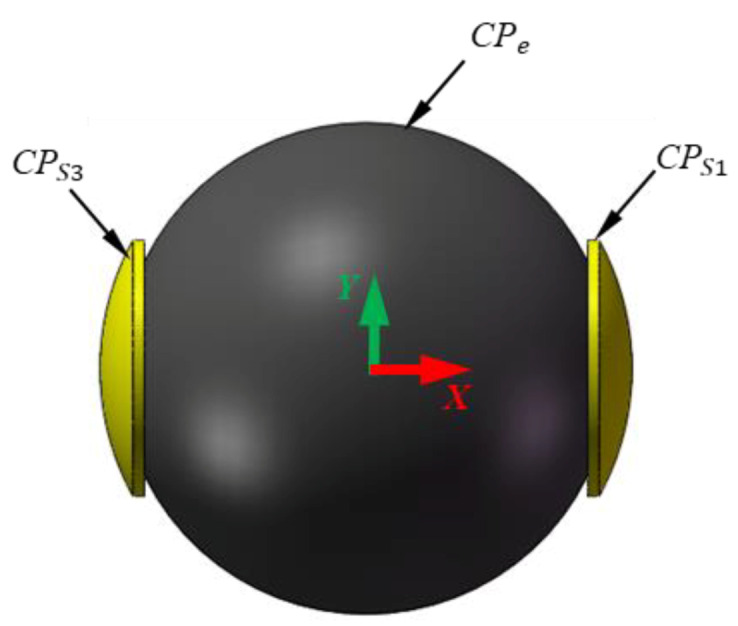
Simulation model of the spherical capacitive sensor.

**Figure 8 micromachines-11-00837-f008:**
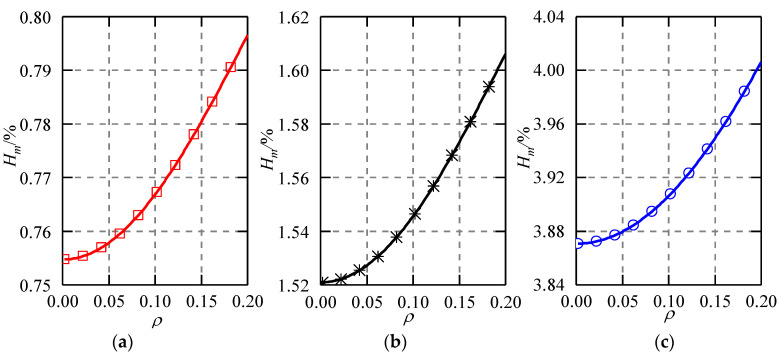
Maximum attitude error caused by shape deviation of the ball. (**a**) Δ*r* = 1 μm, (**b**) Δ*r* = 2 μm, (**c**) Δ*r* = 5 μm.

**Figure 9 micromachines-11-00837-f009:**
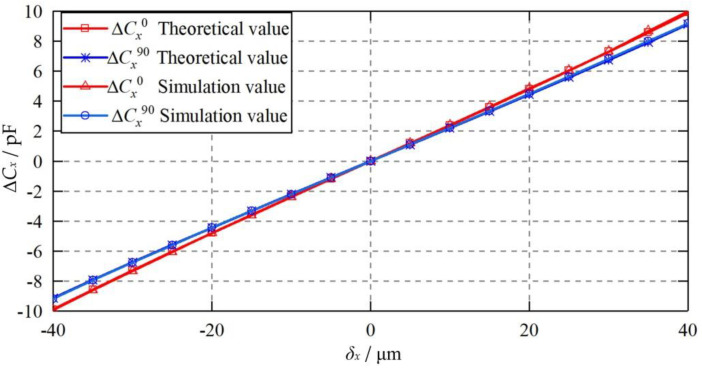
Dependence of the differential capacitance Δ*C_x_* on the eccentric displacement of the ellipsoidal ball.

**Figure 10 micromachines-11-00837-f010:**
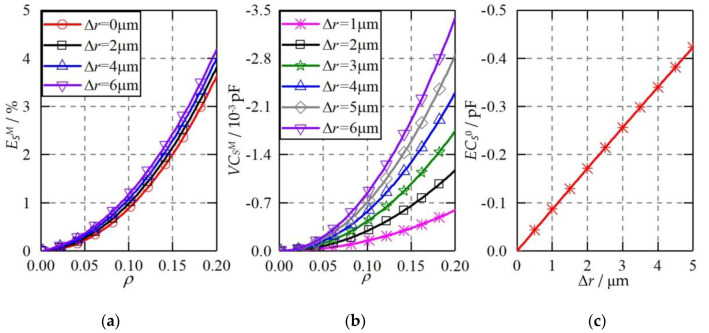
The theoretical capacitance error when the plate shape becomes an ellipsoid. (**a**) Higher-order capacitance error, (**b**) variable capacitance error, (**c**) constant capacitance error.

**Figure 11 micromachines-11-00837-f011:**
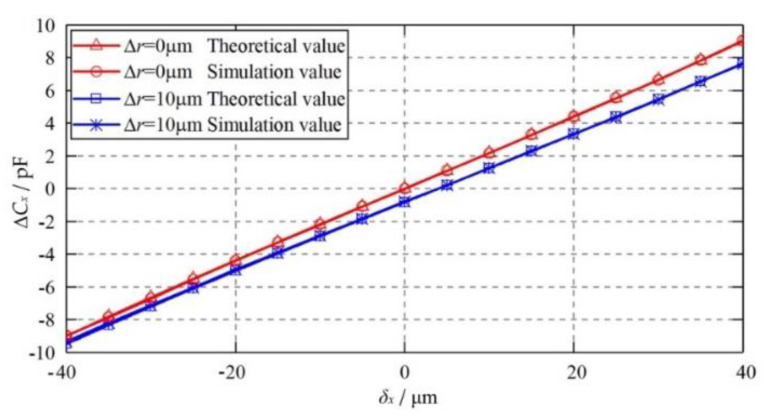
Dependence of the differential capacitance Δ*C_x_* on the eccentric displacement of the ball when the plate becomes an ellipsoid.

**Figure 12 micromachines-11-00837-f012:**
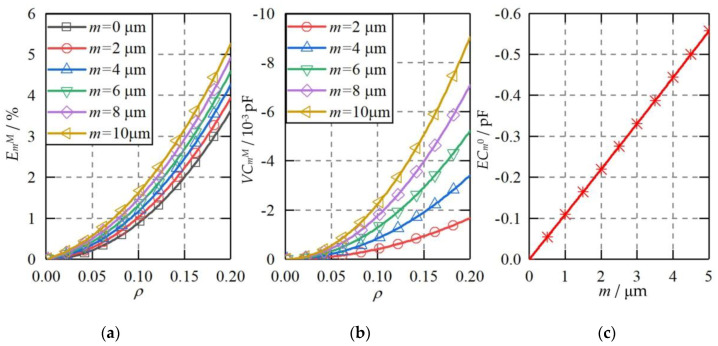
The theoretical capacitance error when the plate is axial offset from the ideal position. (**a**) Higher-order capacitance error, (**b**) variable capacitance error, (**c**) constant capacitance error.

**Figure 13 micromachines-11-00837-f013:**
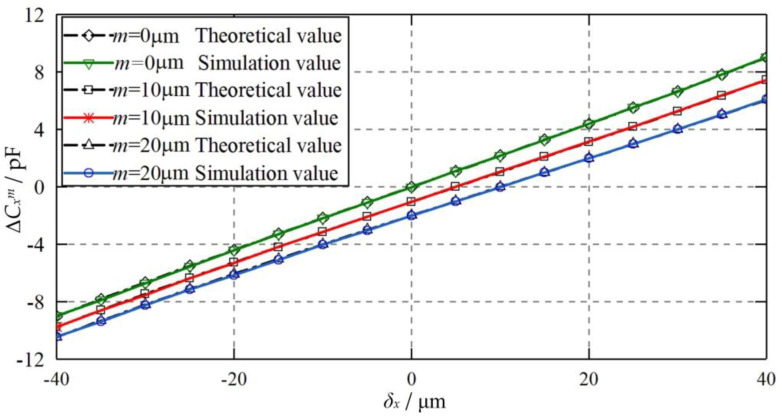
Dependence of the differential capacitance Δ*C_x_^m^* on the eccentric displacement of the ball when the plate is axial offset.

**Figure 14 micromachines-11-00837-f014:**
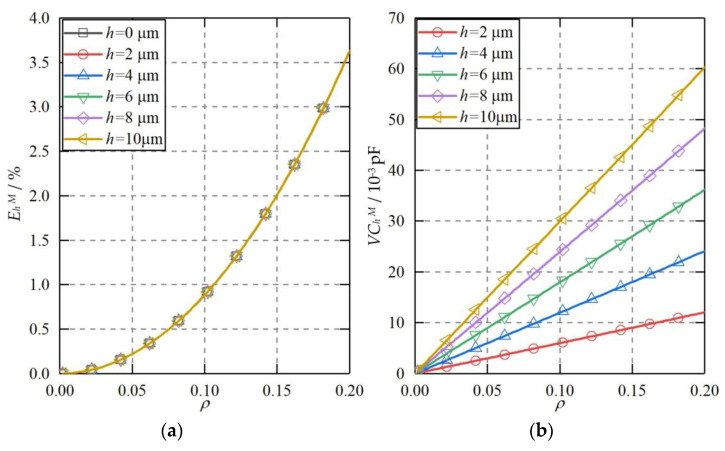
The theoretical capacitance error when *CP_S_*_1_ and *CP_S_*_3_ are misaligned and radial offset (**a**) Higher-order capacitance error, (**b**) variable capacitance error.

**Figure 15 micromachines-11-00837-f015:**
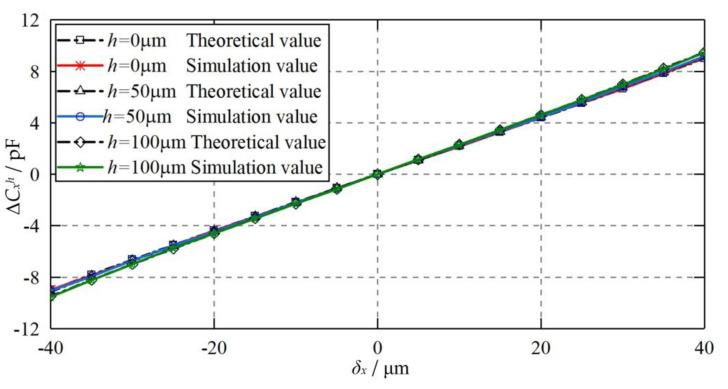
Dependence of the differential capacitance Δ*C_x_^h^* on the eccentric displacement of the ball when *CP_S_*_1_ and *CP_S_*_3_ are radial offset and misaligned with each other.

**Figure 16 micromachines-11-00837-f016:**
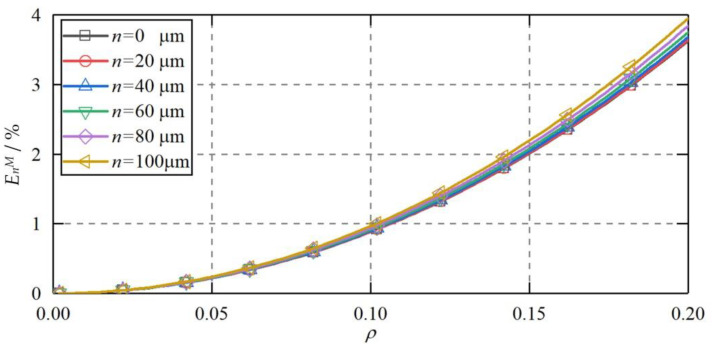
The theoretical capacitance error when *CP_S_*_1_ and *CP_S_*_3_ are radial offset and aligned with each other.

**Figure 17 micromachines-11-00837-f017:**
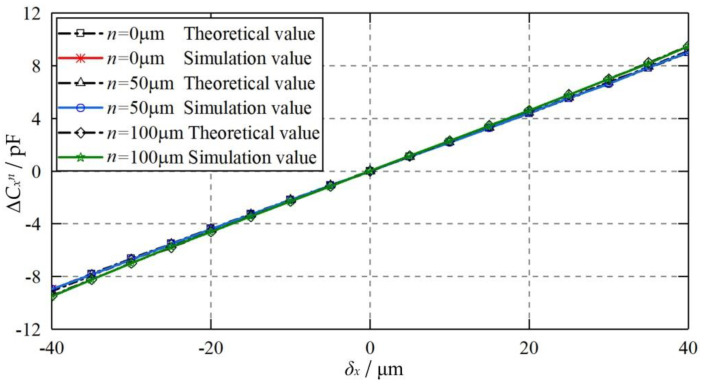
Dependence of the differential capacitance Δ*C_x_^n^* on the eccentric displacement of the ball when *CP_S_*_1_ and *CP_S_*_3_ are radial offset and aligned with each other.

**Figure 18 micromachines-11-00837-f018:**
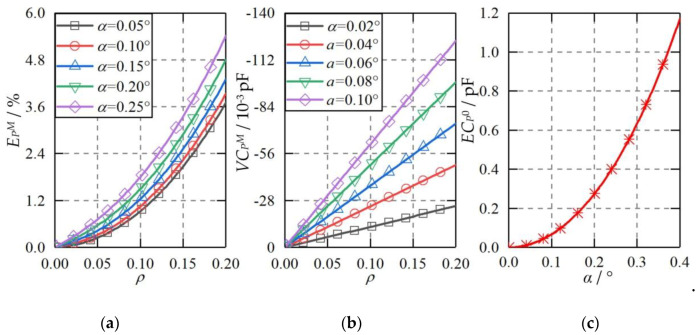
The theoretical capacitance error when *CP_S_*_1_ is inclined with an angle to the ideal position. (**a**) Higher-order capacitance error, (**b**) variable capacitance error, (**c**) constant capacitance error.

**Figure 19 micromachines-11-00837-f019:**
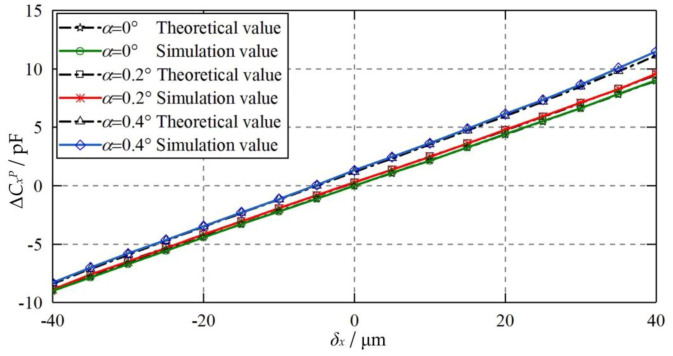
Dependence of the differential capacitance Δ*C_x_^P^* on the eccentric displacement of the ball when *CP_S_*_1_ is inclined with an angle to the ideal position.

**Table 1 micromachines-11-00837-t001:** Parameters of spherical capacitive sensor.

Parameters of Spherical Capacitive Sensor	Value
Central angle of *CP_S_*_1_ ~ *CP_S_*_4_ Central angle of *CP_S_*_5_ and *CP_S_*_6_	0°~30°
	35°~45°
Inner radius of spherical capacitive plate (R)	25 mm
Radius of the ball (r)	24.8 mm

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
