# Peer review of "Error Analysis of a Spherical Capacitive Sensor for the Micro-Clearance Detection in Spherical Joints"

_micromachines, 2020, doi:10.3390/mi11090837_

Round 1

Reviewer 1 Report

The paper describe the error analysis of the spherical capacitive sensor for the detection of micro-clearance. These sensors has many practical application in measurement systems. The authors describe the construction of the sensor. The results of experimental and simulation research are presented. The authors presents evaluation method of metrological properties of sensor. The results are interesting from theoretical and practical point of view.

The paper has good scientific level and is written in good language. The list of References is sufficient.

Below,  a few remarks were presented. Please take them into consideration in final version of this article.

  1. Line 277 and 278: “As shown in figure 5(a), spherical capacitive plate CPS1 is placed away from the ideal position with a distance of h along the positive Y direction while CPS3 along the negative Y direction”. Which Y axis direction is positive?
  2. Line 334: The expression for p1 is illegible. I recommend to change them.
  3. The correct unit of capacitance is pF not PF - Figures 9, 10, 11, 12, 13, 14, 15, 17, 18, 19. I propose change unit of Y axis.

Author Response

We thank this reviewer for the valuable comments and suggested corrections, which help us improve the manuscript. Our reply as per referee’s comments is given below and the changes that have been made to the text have been marked.

1. Line 277 and 278: “As shown in figure 5(a), spherical capacitive plate CPS1 is placed away from the ideal position with a distance of h along the positive Y direction while CPS3 along the negative Y direction”. Which Y axis direction is positive?

Reply: The Cartesian coordinate system XY is defined in detail. The origin of the coordinate is located at the center of the ball. The positive X-axis is towards the center of the plate CPS1 while the positive Y-axis is perpendicular to the positive X-axis based on the right-hand rule. This information is reflected in Lines 277-279.

2. Line 336: The expression for p1 is illegible. I recommend to change them.

Reply: The redundant brackets in this expression have been removed, as shown in Line 336.

3. The correct unit of capacitance is pF not PF - Figures 9, 10, 11, 12, 13, 14, 15, 17, 18, 19. I propose change unit of Y axis.

Reply: The units of the capacitance in the figures 9-15 and 17-19 have been corrected to pF.

Reviewer 2 Report

The paper presents an error analysis of a spherical capacitive sensor with differential structure and explores the dependence of the differential capacitance on the manufacturing and the installation imperfectness. Five error sources are examined, i.e., the shape of the ball and the capacitive plate, the axial and radial offset of the plate, and the inclined installation of the plate. The mathematical models for calculating the capacitance errors of the spherical capacitive sensor are deduced and validated by the simulation using Ansoft Maxwell. The results show that the measurement accuracy of the spherical capacitive sensor is significantly affected by the shape of plates and ball, the axial offset, and the inclined angle of the plate.

The paper is of interest to readers and I recommend its publication

Author Response

We really thank this reviewer for the valuable comments.